# Animating hydrogel knotbots with topology-invoked self-regulation

Qing Li Zhu [1,4], Weixuan Liu [2,4], Olena Khoruzhenko[3], Josef Breu [3], Wei Hong [2] ✉, Qiang Zheng [1] ✉ & Zi Liang Wu [1] ✉

Steering soft robots in a self-regulated manner remains a grand challenge, which often requires continuous symmetry breaking and recovery steps for persistent motion. Although structural morphology is found significant for robotic functions, geometric topology has rarely been considered and appreciated. Here we demonstrate a series of knotbots, namely hydrogel-based robots with knotted structures, capable of autonomous rolling and spinning/rotating motions. With symmetry broken by external stimuli and restored by self-regulation, the coupling between self-constraint-induced prestress and photothermal strain animates the knotbots continuously. Experiments and simulations reveal that nonequilibrium processes are regulated dynamically and cooperatively by self-constraints, active deformations, and self-shadowing effect of the photo-responsive gel. The active motions enable the knotbots to execute tasks including gear rotation and rod climbing. This work paves the way to devise advanced soft robots with self-regulated sustainable motions by harnessing the topology.

Directional motions in nature, ranging from molecular motors to running cheetahs, often rely on structural asymmetry and self-regulation mechanisms[1–3], with feedback through internal and/or external pathways. For example, proteins excel in receiving electro-chemical signals, converting conformational asymmetry into directional motion, and restoring their configurations autonomously through biochemical feedback loops[4]. As another example, some multicellular organisms exhibit highly cooperative motions by symmetry breaking through the interactions between the body and the environment[1,3]. These biological systems provide inspiration for the design of artificial soft robots; however, it remains a grand challenge to replicate the sophisticated actuation and control systems of biological systems with intricate responses to bio-signals and complex neural networks. For the design of continuum robots, soft active materials have been exploited to perform continuous motions based on symmetry breaking and external stimulations which often require complex manual control[5–10]. In these artificial systems, external or internal feedback loops are incorporated[11–13], such as self-shadowing effect[14–16] or oscillatory chemical reactions[17,18]. Although similar mechanisms have been employed to drive artificial molecular motors by introducing steric hindrance or structural chirality[19,20], self-regulated symmetry restoration has seldom been realized in continuum soft robots. It is even more challenging to achieve self-regulated continuous motions under time-invariant conditions. As a possible solution, specially designed morphologies that are invariant against desired modes of motion (e.g., rolling and rotation), i.e., a type of temporal self-similarity[21], can be engineered into continuum robots to accomplish self-regulated smooth motion.

Typical symmetry-breaking strategies include asymmetric shapes[22,23], gradient structures, and asymmetric external stimuli[9,24,25]. However, topology-based symmetry restoration has rarely been explored in soft actuators or robots. According to knot theory[20,26],

[1]Ministry of Education Key Laboratory of Macromolecular Synthesis and Functionalization, Department of Polymer Science and Engineering, Zhejiang University, 310058 Hangzhou, China. [2]Shenzhen Key Laboratory of Soft Mechanics & Smart Manufacturing, Department of Mechanics and Aerospace Engineering, Southern University of Science and Technology, 518055 Shenzhen, China. [3]Bavarian Polymer Institute and Department of Chemistry, University of Bayreuth, Universitätsstrasse 30, 95440 Bayreuth, Germany. [4]These authors contributed equally: Qing Li Zhu, Weixuan Liu. ✉e-mail: hongw@sustech.edu.cn; zhengqiang@zju.edu.cn; wuziliang@zju.edu.cn

knots are a type of topological element with invariant morphologies and closed curves in three-dimensional space. Knots are more often used in daily lives than engineering and can be used to break various types of symmetry[27,28], such as mirror symmetry, axial symmetry, and chiral symmetry, thus allowing for flexible gait control in an engineered locomotor system. In addition, owing to the interlacement of closed loops, knots often include topological constraints, bring self-shading and prestress, and thus may enhance actuation performance and afford physical intelligence (i.e., capacities such as adaptation, actuation, and/or control physically encoded in the soft body of an agent)[5,29,30]. When all these means of symmetry breaking/restoration are combined with the responsiveness of active materials, a promising roadmap in designing soft robots with self-regulated motile performance can be foreseen.

Here we present a series of soft robots with knotted structures, the *knotbots*, which are made from a photo-responsive anisotropic hydrogel and capable of autonomous and continuous motions upon light stimulation. By breaking the rotational symmetry, mirror symmetry, spatiotemporal symmetry, and/or chiral symmetry, various types of light-activated knotbots are designed to achieve overall spinning/rotation as well as local rolling, through the coupling between self-constraining geometry and photothermal stress. Sustained motion is guaranteed by the special self-similar morphologies invariant to the spinning, rotation, and rolling motions. Various knotbot designs, including torus, single prime knots (e.g., trefoil and pentafoil knots), and complex links (e.g., Solomon and Star-of-David links), have been illustrated. Influences of other parameters, ranging from the size and chirality of the knots to the intensity and relative direction of the light, on the motile behaviors of the continuum robots are also examined. Through theoretical models, it is further confirmed that such regulated motions originate from the synergistic contributions by the geometry-induced prestrain, stimulus-actuated shape change, and the self-shadowing effect. Furthermore, it is demonstrated that the dynamic motions of knotbots can be used to perform other complex tasks, such as rotating gears and climbing a rod under uniform light irradiation, and could bring potential impacts to diverse fields.

## Results

### Synthesis, structure, and response of cylindrical gels

Fluorohectorite $[Na_{0.5}][Li_{0.5}Mg_{2.5}][Si_4]O_{10}F_2$ nanosheets (NSs) and gold nanoparticles (AuNPs) are incorporated into the precursor solution to prepare responsive, anisotropic poly(N-isopropylacrylamide) (PNIPAm) hydrogel. The aqueous suspensions of NSs with high charge density of $1.1\,nm^{-2}$ and aspect ratio of ~20,000 exhibit nematic phase at a very low content of NSs[31], which are easily oriented by mechanical shear[32] (Supplementary Fig. 1). Anisotropic hydrogel noodles are prepared by polymerizing the precursor solution after brisk injection into a silicone tube (Fig. 1a). The gel shows strong birefringence under polarizing optical microscope (POM), while the cross-section exhibits a Maltase cross, indicating concentric alignments of NSs within the cylinder (Fig. 1b). Shear-induced alignment of NSs in the suspension can be maintained for at least 10 min, despite a slight relaxation (Supplementary Fig. 2), allowing photopolymerization to stabilize the anisotropic structure. After systematic experiments, the synthesis process is optimized with a flow rate of $10.6\,mm\,s^{-1}$ and NS content of 1.0 wt% (Supplementary Figs. 3 and 4). The alignment of NSs and presence of AuNPs are confirmed by scanning and transmission electron microscopy (SEM and TEM) (Fig. 1c and Supplementary Fig. 5). The alignment of NSs is also confirmed by small-angle X-ray scattering (SAXS) measurements (Fig. 1d, e). An array of scattering spots indicates that the spacing between cofacially oriented NSs in the as-prepared gel is ~36.7 nm (Supplementary Fig. 6 and Supplementary Table 1). After equilibrated in water, the hydrogel maintains the anisotropic structure of NSs with an orientation degree of 0.84.

The nanocomposite PNIPAm hydrogel exhibits anisotropic deformation when subject to heating or light irradiation. The gel readily contracts along the axis and slightly expands in the radial direction, after being transferred from a 25 to 40 °C water bath (Supplementary Fig. 7). This fast, anisotropic deformation of the gel is related to the dehydration of PNIPAm chains when heated above the low critical solution temperature (LCST ≈ 32 °C); the release of water molecules previously bound to polymer chains results in a sudden increase in permittivity and thus enhances the electrostatic repulsion between the highly charged NSs[22,33]. Cyclic switch between 25 and 40 °C water baths leads to repeated, fast, and isochoric deformation of the gel. To correlate the macroscopic deformation of the gel with the varied alignment of NSs, SAXS measurements are performed on the gel during the heating-cooling process. When heated from 25 to 40 °C, the anisotropy of ellipsoidal scattering gradually decreases, corresponding to the reduced orientation degree of NSs from 0.84 to 0.52 (Fig. 1d, e and Supplementary Fig. 8). When the gel is cooled down to 25 °C, the scattering pattern recovers the original.

As a high-efficiency photothermal transduction agent[14,34], AuNPs afford the gel with response to light (Supplementary Fig. 9). Upon light irradiation (wavelength, 520 nm; intensity, $0.8\,W\,cm^{-2}$), the local temperature of the gel rises from 25 to 40 °C in 3 s. Due to the limited depth of light penetration, the lateral gradient of thermal-strain bends the cylindrical gel with an amplitude of 60° after light irradiation for 3 s. The gel recovers its initial shape in ~10 s after removal of the light (Fig. 1f). Therefore, repeated light irradiation results in fast and reversible bending of the gel (Fig. 1g and Supplementary Movie 1). Limited by the light penetration and heat transfer, the peak temperatures on the sides facing and dorsal to the light source differ by 10 °C (Fig. 1h). As expected, the bending speed and amplitude of the gel are related to the composition and diameter of the cylindrical gel, as well as the AuNP content and the power intensity. After considering the mechanical and responsive behaviors (Supplementary Figs. 10 and 11), the cylinder gel with NS content of 1.0 wt%, AuNP content of 0.38 wt%, and diameter of 2 mm is selected to devise the soft robots.

### Motions of gel tori under static and dynamic light

The cylindrical hydrogel is bent and jointed at the ends without twist. The resulting torus is also known as a topologically trivial knot[26], which is denoted as $T_0$ knotbot. This initial geometry introduces a field of prestrain−compressive on the inner side and tensile on the outer side (Fig. 2a)−and its distribution is invariant against rolling[21]. Light irradiation atop the torus leads to a vertical temperature gradient and thus thermal-strain mismatch in the cross-section−the irradiated top tends to shrink and roll inward to compensate for this mismatch after the mirror symmetry is broken between the top and bottom of the torus. The rolling motion of the torus under light irradiation can be understood by considering the hoop stresses (i.e., circumferential stresses imposed on the cylindrical gel)−the top half is subject to tension while the bottom is to compression (Fig. 2b and Supplementary Fig. 12a), in addition to the prestress. The tensile and compressive hoop stresses yield concentric and eccentric resultant forces, respectively, and the separated action lines induce a couple of moment, which drives the inward rolling of the torus[24]. It should be noted that self-similarity is maintained in the torus while rolling, which is important for continuous motion.

The rolling motion is characterized by the rolling angle, $\gamma$, about the centroidal axis of the torus (Fig. 2b). Under uniform light irradiation with appropriate intensity $I$ (e.g., $0.8\,W\,cm^{-2}$), the distributed couple moment leads to autonomous rolling of the $T_0$ knotbot at a constant speed $\dot{\gamma}$ ($6.8°\,s^{-1}$) after a short acceleration stage (Fig. 2c, d and Supplementary Movie 2). The rolling motion of the torus is closely related to environmental temperature and power intensity of light, which determine the steady state of heat conduction and cooling to ensure self-adapted morphing and continuous motion (Fig. 2e). Too

 

low power intensity and environmental temperature cannot provide sufficient moment to drive the rolling motion, while too high intensity of light and temperature of bath lead to gradual contraction of gel torus and final ceasing of the motion due to insufficient cooling (Supplementary Fig. 12). The rolling motion highly relies on the temperature gradient, in other words, the destruction of mirror symmetry in the vertical direction. The rolling of a $T_0$ knotbot tends to tilt the

vertical temperature gradient horizontally to restore symmetry. However, the light irradiation constantly installs the vertical temperature gradient against restoration. Guaranteed by self-similarity during rolling, the continuous destruction and restoration of symmetry lead to the sustained motion of the knotbot. Continuous rolling is found in the gel torus over wide-spectrum conditions, indicating the robustness of the autonomous motion.

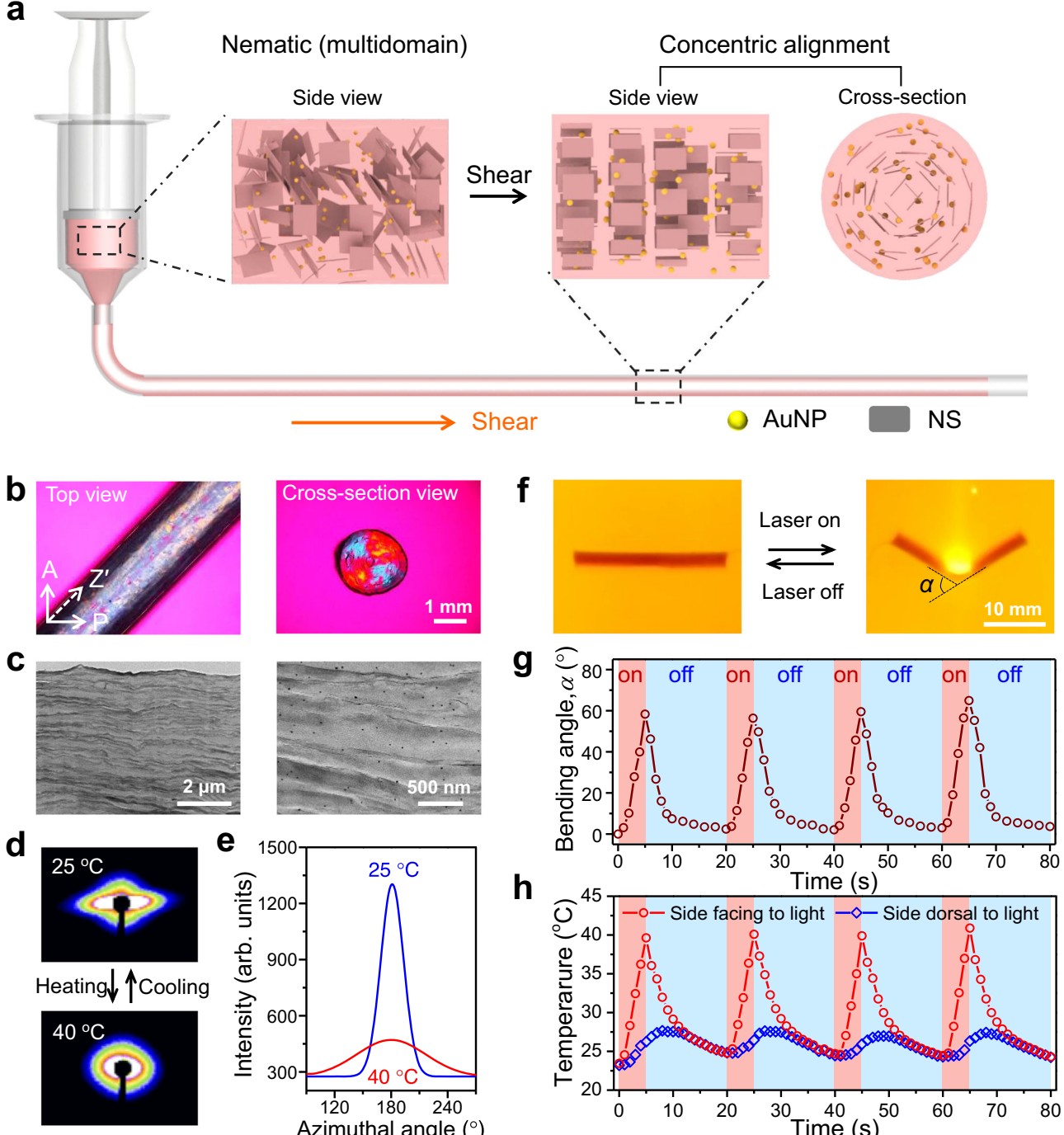

**Fig. 1 | Synthesis, anisotropic structure, and the response of cylindrical hydrogel to temperature and light. a** Synthesis of anisotropic gel with concentric alignments of NSs by shear orientation followed by photopolymerization. For simplicity, monomer, initiator, and chemical crosslinker are omitted in the scheme. **b** POM images of the anisotropic gel. A: analyzer; P: polarizer; Z': slow axis of 530 nm tint plate. Representative images of $n$ = 3. **c** TEM images of the cross-section of the anisotropic gel. The black dots are AuNPs in the gel. Representative

images of $n$ = 3. **d, e** SAXS patterns (**d**) and intensity-azimuthal plots (**e**) of the gel during heating and cooling. **f** Photos of light-induced bending and unbending of the gel. **g** Variation of bending angle of the cylindrical gel upon cyclic irradiation of a light spot. **h** Variations of local temperatures of the gel on the sides facing and dorsal to the light upon cyclic irradiation. Diameter of the gel, 2 mm; power intensity, 0.8 W cm².

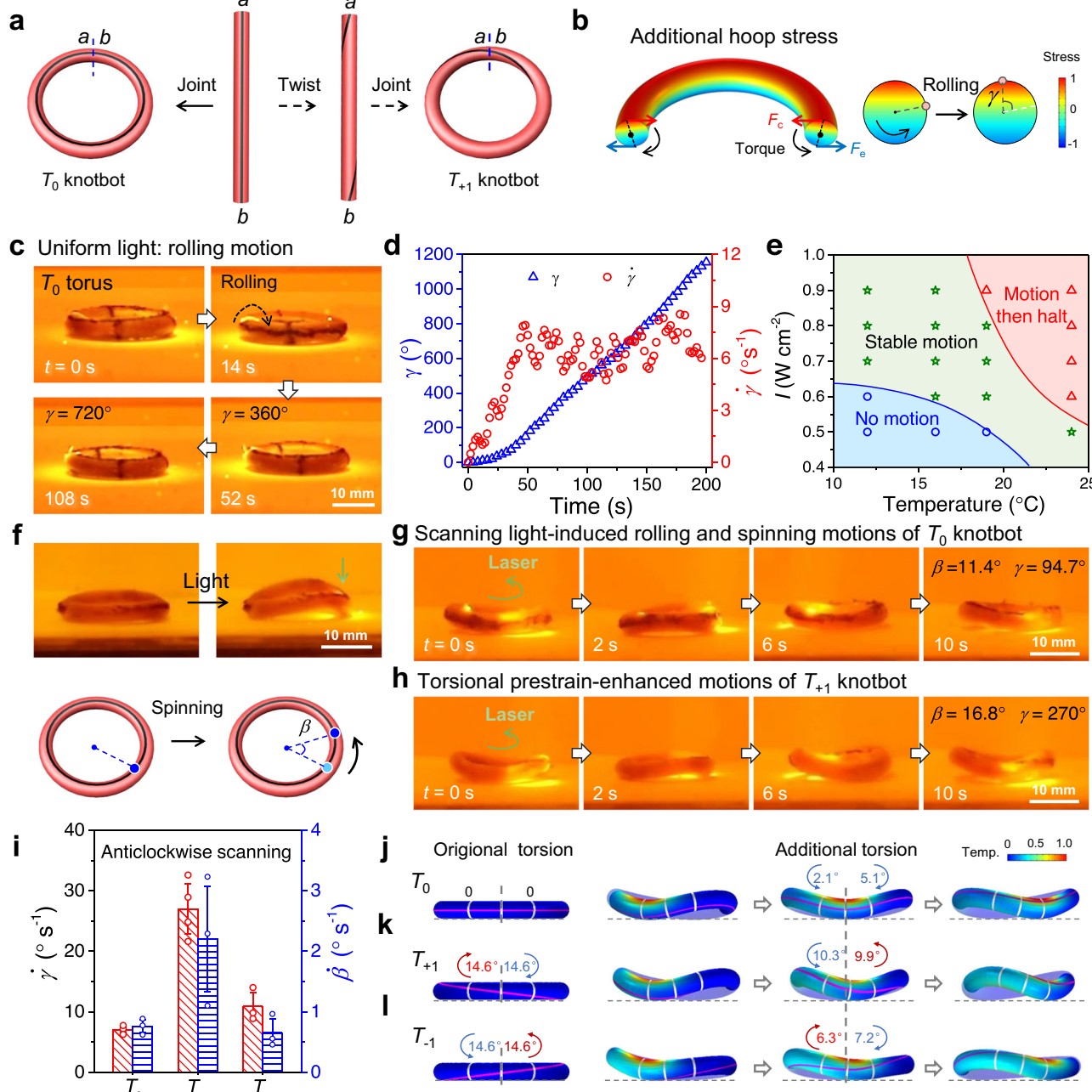

**Fig. 2 | Light-steered continuous spinning and rolling of hydrogel knotbots.**
**a** Schematics for the preparation of knotbots with and without torsional prestrain.
**b** Simulated excess hoop stress distribution in a $T_0$ knotbot due to uniform light irradiation. $F_c$ and $F_e$ represent the resultant concentric force and eccentric force, respectively. The resultant hoop stress yields a couple moment in the cross-sectional plane, causing inward rolling with the rolling angle denoted as $\gamma$. **c** Photos of continuous motion of a $T_0$ knotbot irradiated under uniform light. **d** Rolling angle $\gamma$ and rolling speed $\dot{\gamma}$ as functions of irradiation time. Environmental temperature, 19 °C; power intensity, 0.8 W cm$^{-2}$. **e** Phase diagram of the motion state of a $T_0$ knotbot in a water bath with different temperatures and under uniform light with different intensities. **f** Photos and schematics of the spinning motion of a $T_0$ knotbot under scanning-light irradiation. The spinning angle is denoted as $\beta$. **g, h** Photos of continuous anticlockwise spinning and inward rolling of $T_0$ (g) and $T_{+1}$ (h) knotbots under anticlockwise scanning light. **i** Spinning speed $\dot{\beta}$ and rolling speed $\dot{\gamma}$ of $T_0$, $T_{+1}$, and $T_{-1}$ knotbots under identical anticlockwise scanning light. **j–l** Numerical results of deformed shapes and motion kinematics of $T_0$ (j), $T_{+1}$ (k), and $T_{-1}$ (l) knotbots under anticlockwise scanning light. The additional torsion caused by localized photo-heating is marked on the neighboring regions of the torus, with the cross-section under the light spot taken as a reference. The knotbots with torsional prestrains have large differences in additional torsions and contact points. Data points are represented as mean ± s.d. ($n \geq 3$).

Another strategy for continuous motion is to break the temporal symmetry of the stimuli. As shown in Fig. 2f, continuous rolling is also achieved by scanning a light spot with a high intensity of 1.2 W cm$^{-2}$ circumferentially along the gel torus, so that each section is exposed intermittently, leaving ample time for cooling down. For example, the light scanning at a speed of 5 mm s$^{-1}$ results in continuous rolling at a speed $\dot{\gamma}$ of 7.0° s$^{-1}$. The dynamic scanning of light also deforms the torus into a shape that breaks both the rotational and mirror symmetries. The region exposed to light locally bends upward and warps the torus out of the plane, forming a saddle shape in contact with the supporting surface only at two sections. The warped region travels along with the scanning-light spot, leading to the continuous switching of contact points along the circumferential direction[35] (Fig. 2g), which forces the knotbot to move in the same direction that locally

resembles the rolling contact of a wheel. This traveling deformation also leads to the spinning of the torus knotbot[6,25], which is characterized by the spinning angle $\beta$ of its cross-section about the revolution axis (Fig. 2f). The $T_0$ knotbot spins slowly at a speed $\dot{\beta}$ of -1.1° s$^{-1}$ (Supplementary Movie 3). It should be noted that the prestrain, which flattens the energy landscape for rolling, plays a significant role in the motion. A directly molded gel torus without prestrain fails to overcome the energy barrier against flipping through and shows only traveling bending under the scanning light (Supplementary Fig. 13).

To further break the symmetry and enhance the actuation performance, we incorporate internal chirality into the torus knotbot by twisting the cylindrical gel before joining the two ends (Fig. 2a). As a shorthand, a gel twisted clockwise by 360° with right-handed chirality is noted as the $T_{+1}$ knotbot. As shown in Fig. 2g, a $T_{+1}$ knotbot exhibits continuous spinning at a speed $\dot{\beta} \approx 1.6°$ s$^{-1}$ and rolling at a speed $\dot{\gamma} \approx 25.9°$ s$^{-1}$ (Supplementary Movie 3), markedly faster than a non-chiral $T_0$ knotbot under the same condition. Moreover, under anticlockwise light-scanning atop, the $T_{-1}$ knotbot exhibits slower rolling and spinning than the $T_{+1}$ knotbot (Fig. 2h). Numerical models are formulated to reveal detailed kinematics. It is noteworthy that, due to the asymmetry between heating and cooling, the scanning-light-induced warping also brings local chirality to a non-chiral $T_0$ knotbot, causing its directional motion. For a knotbot with pre-torsion, the enhancement in actuation depends on the superposition between the two chiral contributions (Fig. 2j-l). When a $T_{+1}$ knotbot is subject to anticlockwise light scanning, the pre-torsion and the sequence of scanning act constructively to the asymmetry in the warped shape between the two sides of the contact point[36], thus improving the locomotion. Whereas, in the $T_{-1}$ knotbot, anticlockwise light scanning is destructive to the effect of pre-torsional strain and therefore reduces the spinning and rolling speeds (Supplementary Fig. 14). As the main purpose of the numerical simulations is to illustrate the governing mechanism, simplifications are employed to facilitate calculation, and qualitative agreement is expected. Quantitative predictions may be possible with extended models but are beyond the scope of the current paper (Supplementary discussions of theoretical modeling).

The spinning and rolling speeds of a torus knotbot depend on its size, pre-torsion, and power intensity. The increase in the pre-torsion from 0 to 540° ($T_0$ to $T_{+1.5}$) boosts both the spinning and rolling speeds of the knotbot under anticlockwise scanning of light atop (Supplementary Fig. 15a). Too much pre-torsion (e.g., $T_{+2}$ knotbot), however, may lead to Michell's instability which deforms the torus into a figure-of-eight shape[29] and disrupt the actuation mechanism (Supplementary Fig. 16). On the other hand, the power intensity influences the motion of a knotbot through the amplitude of photothermal strain and the balance with effective cooling (Supplementary Fig. 15b). For a similar reason, the size of a gel torus also affects the motion, as detailed in theoretical modeling and Supplementary Fig. 15c. Although the motile behaviors of knotbots are sensitive to experimental parameters, the motions exhibit good repeatability and reliability at identical conditions. The sensitivity of motions to variables also provides avenues to tune the motile behaviors of the knotbots.

## Rolling and rotation of trefoil knotbots

The trefoil knot, which has three crossings and intrinsic geometric chirality, is the simplest prime knot according to the knot theory[26]. As shown in Fig. 3a, the cylindrical gel is manually tangled into a right-handed trefoil knot before jointed at the two ends. The structural topology endows the trefoil knot with non-uniform prestrain and geometric chirality, breaking the mirror symmetries in all directions, and reducing the axial symmetry to a 3-fold rotational symmetry[27]. The reduced symmetry and presence of crossings invigorate a specific mode of motion, which will be referred to as the braid rotation hereafter. With one strand rolling on top of the other at a crossing, the hoop stresses in a single strand also induce inward rolling. When the

entire cross-section of both strands is taken into account, the same mechanism drives the rotation of the entire braid, i.e., braid rotation. Accordingly, the trefoil knotbot exhibits a gradual change in the relative positions of the crossings and an apparent rotation of the knot (Fig. 3b). This braid rotation will be characterized by the rotation angle $\theta$ of a crossing. Because of the self-shadowing effect at crossings and the self-adapted deformation of the knotbot, under uniform light irradiation (intensity, 0.8 W cm$^{-2}$), a right-handed trefoil knotbot exhibits autonomous and continuous inward rolling at a speed $\dot{\gamma}$ of 12.4° s$^{-1}$ and anticlockwise rotation at a speed $\dot{\theta}$ of 7.8° s$^{-1}$, after a short starting period (Fig. 3c and Supplementary Movie 4).

To reveal the underneath kinematics, we examine the temperature and stress distributions in the trefoil knotbot during light stimulation (Fig. 3d, e and Supplementary Fig. 18). Just as in a torus, the hoop stresses on the cross-section generate a couple moment which tends to roll the segment inward. Because of the shadowing effect, the segment above each crossing is irradiated and thus heated up more than the one underneath. When both strands in the braid are regarded as a cross-section, the couple moment generated by the distributed hoop stress also rotates the braided strand, leading to the apparent profile rotation (Fig. 3d). Due to friction, the contact points at the bottom serve as the anchor points of the composite motion, and the continuous shift in the contact points determines the relative speed between the material-particle motion and braid rotation. For the trefoil knotbot tested, the material-particle motion is relatively slow and mostly reciprocating, as shown by the grid marked on the knotbot (Fig. 3c, e), while the apparent rotation is perceivable from the motion of petals and crossings, with the profile rotation as the dominant component. The three modes of motion from simulation results are shown in Fig. 3d, f. The rolling angle $\gamma$ is obtained by tracking a material particle, while the spinning and braid rotation are indicated by the average position of a specific cross-section change in polar coordinates. The periodic fluctuations in both polar angle $\phi$ and polar radius $\rho$ are caused by braid rotation, while the downtrend of $\phi$ manifests the insignificant spinning motion, which is induced by the asymmetry friction of the trefoil knot against the ground. In this trefoil knotbot, the topological structure and constraint result in self-regulation and high cooperativity of these motions under uniform light stimulation.

It has also been noticed that the overall size of a trefoil knotbot, relative to the strand diameter, plays a crucial role in the modes of motion. When the same gel noodle is tied into a smaller knot, the motion is hampered by the tighter constraint and thus higher friction[37], and a smaller window for cooling. To achieve effective actuation, a properly sized trefoil knotbot is irradiated under uniform light. As shown in Fig. 3g, after 60 s, $\theta$ rises to 425° and $\gamma$ accumulates to 677° at a light intensity of 0.8 W cm$^{-2}$. Experiments show that $\theta$ is linear in $\gamma$ (Fig. 3h), indicating that rotation is a direct consequence of rolling. For the same trefoil knot, it is noticed that the ratio of $\theta$ to $\gamma$ is constant under uniform light of different intensities. Systematic investigation reveals that continuous motion of a trefoil knotbot can be achieved when a dynamic equilibrium is reached between heating and energy dissipation (Fig. 3i). The rotation direction depends on the chirality of a knotbot, relative to the direction of light irradiation. Under uniform light irradiation from the top, a left-handed trefoil knotbot rotates clockwise (Supplementary Fig. 19a). When irradiated from the bottom, opposite directions will be taken by the same knotbot in both rolling and rotation, demonstrating the switchability in actuation (Supplementary Fig. 19b).

Besides the structural chirality, a scanning light may also be used to introduce spatiotemporal chirality to animate a trefoil knotbot. Similar modes of motion are observed—a properly sized trefoil knotbot demonstrates continuous rolling and braid rotation. As shown in Fig. 3j, after one cycle of clockwise light scanning in ~10 s (power intensity, 1.2 W cm$^{-2}$), the right-handed knotbot rolls inwards by 360°, and rotates anticlockwise by 265° (Supplementary Movie 5). Evidently,

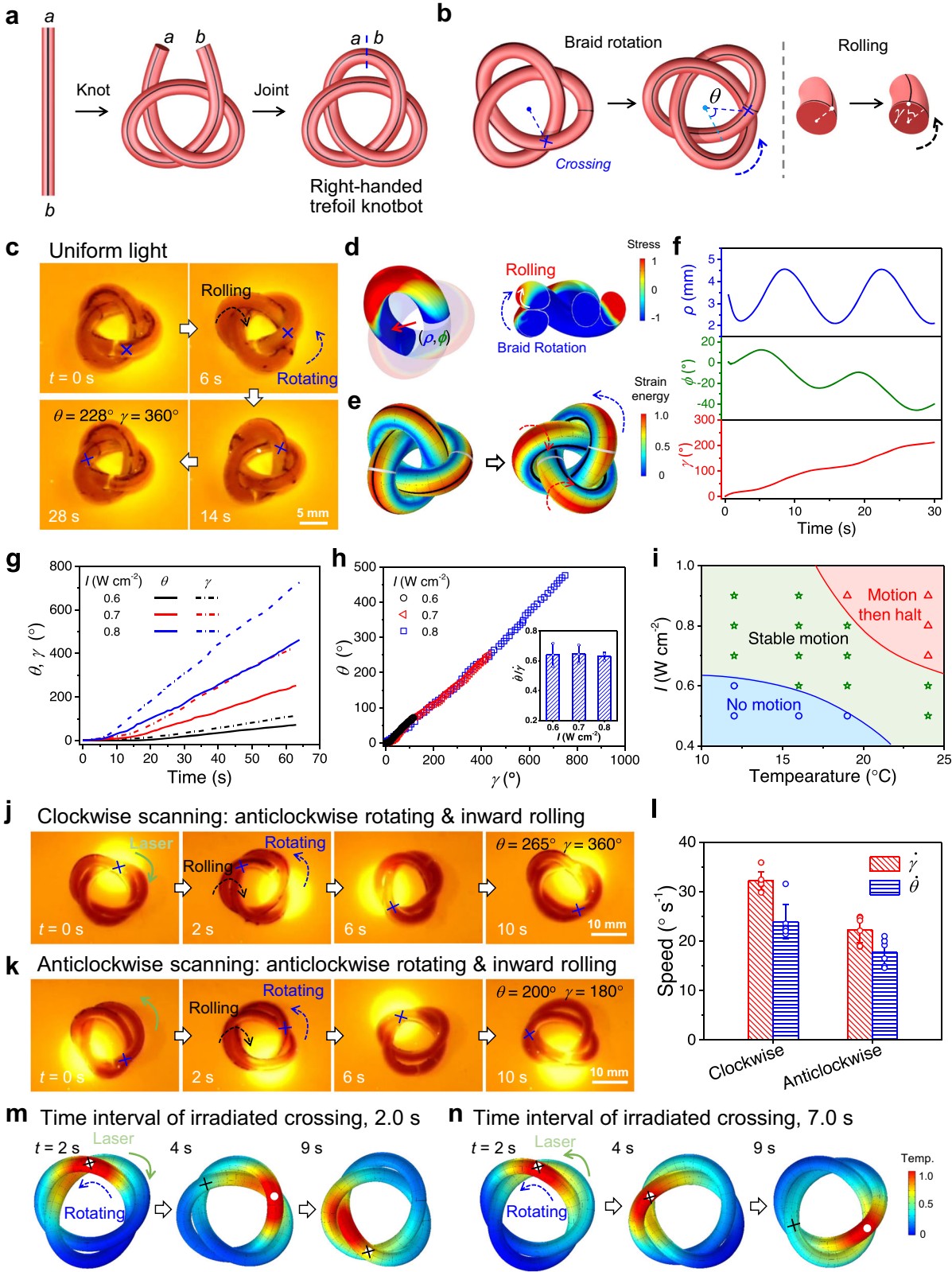

the effect of structural chirality is much more prominent than that introduced by scanning light. A right-handed trefoil exhibits anticlockwise rotation, regardless of the light-scanning direction, although faster rotation is observed when the two effects contribute synergistically. For example, anticlockwise light-scanning atop the same right-handed trefoil knotbot significantly slows down both modes of motion (180° rolling and 200° rotation for each cycle) (Fig. 3k, l and

Supplementary Movie 5). The detailed kinematics is further revealed by numerical simulations. Due to the shading effect, crossings are the active units for braid rotation, and thus the frequency of crossings irradiated by the scanning-light spot influences the motion speeds of the knotbot. As shown in Fig. 3m, n, under scanning along different directions, the crossings are irradiated at different frequencies. Since the crossings of trefoil knotbot rotate anticlockwise, opposite to the

**Fig. 3 | Autonomous rolling and rotation of trefoil knotbots upon light irradiation. a** Preparation of a right-handed trefoil knotbot. **b** Schematics of rolling and braid rotation of a trefoil knotbot. **c** Anticlockwise rotation and inward rolling of a trefoil knotbot under uniform light irradiation from the top. **d, e** Simulated stress distribution in a trefoil knotbot: (d) additional hoop stress with dynamic motion and (e) von Mises stress under uniform light irradiation. **f** Simulated results of polar angle $\phi$ and polar radius $\rho$ of a specific cross-section, as well as the rolling angle $\gamma$ of a specific particle, as functions of time under uniform light irradiation. **g** Variations of rotation angle $\theta$ and rolling angle $\gamma$ of the trefoil knotbot as functions of time under uniform light with different intensities. **h** Variation of $\theta$ as a function of $\gamma$. The

inset presents the ratio of rotation speed $\dot{\theta}$ to rolling speed $\dot{\gamma}$ of the knotbot. **i** Motion state diagram of the trefoil knotbot in water bath with different temperatures and under uniform light with different intensities. **j, k** Rolling and braid rotation of a right-handed trefoil knotbot upon clockwise (j) and anticlockwise (k) scanning light. **l** $\dot{\theta}$ and $\dot{\gamma}$ of the right-handed trefoil knotbot under clockwise and anticlockwise scanning light. **m, n** Simulations and schematics illustrating the kinematics of a right-handed trefoil knotbot with different motion speeds under clockwise (m) or anticlockwise (n) scanning of light. Data points are represented as mean ± s.d. ($n \geq 3$).

clockwise scanning light, the superposition enables more frequent animation of the crossings and thus accelerates the knotbot. Experiments and simulations show that the average interval between two subsequent irradiations of a crossing is 2 s under clockwise scanning and 7 s under clockwise scanning. Such a picture in Fig. 3j, k qualitatively illustrates the dynamic interaction between the structural chirality from topology and the spatiotemporal chirality induced by the stimulation sequence. Just like in a torus knotbot, although the prestrain introduced at the time of fabrication is not a driving force for the motion, it facilitates the dynamic process by flattening out the energy barriers against rolling. As a reference, a directly cast trefoil gel, without any prestrain, does not exhibit rolling or spinning (Supplementary Fig. 20).

## Universality and functionalities of gel knotbots

The design principle illustrated above can be generalized to devise knotbots of diverse shapes and topologies. We have prepared pentafoil knotbots with specific chirality that exhibit similar rolling and rotation under light stimulation, indicating comparable kinematics (Fig. 4a, Supplementary Fig. 21, and Supplementary Movie 6). The crossings of knotbots facilitate the self-shadowing effect to regulate the local deformation for continuous rotation. This mechanism is also manifested in a Hopf-link knotbot with two interlocked tori (Supplementary Fig. 22). Moreover, a right-handed Solomon link with four crossings is fabricated with two intertwined gel strands, which shows autonomous and continuous inward rolling at a speed of $9.1° \, s^{-1}$ and anticlockwise rotation at a speed of $4.5° \, s^{-1}$ under uniform light irradiation from the top (Fig. 4b, d and Supplementary Movie 6). Like a trefoil knotbot, the Solomon-link knotbot also exhibits continuous braid rotation and rolling under scanning light with spatiotemporal symmetry breaking (Supplementary Fig. 23). Torsional prestrain can also be introduced into a Solomon-link knotbot to break the symmetry further. The two hydrogel strands are anticlockwise (or clockwise) twisted by 360° and joined together to form a right-handed Solomon link, which is denoted as an $SL_{+1}$ ($SL_{-1}$ for left-handed) knotbot. Under anticlockwise scanning of light, the $SL_{+1}$ knotbot exhibits faster rolling and rotation than the $SL_{-1}$ knotbot, due to the coupling between the photothermal actuation and the torsional prestrains, similar to the results of torus knotbots. As expected, under clockwise scanning light, the rolling and rotation of the $SL_{-1}$ knotbot are faster than the $SL_{+1}$ knotbot (Supplementary Fig. 24). Similar phenomena are observed in a Star-of-David link knotbot having six crossings of two interwoven gel tori, which is actuated by the same mechanism. As shown in Fig. 4c, Supplementary Fig. 25, and Supplementary Movie 6, under uniform light irradiation from the top, the Star-of-David link knotbot exhibits autonomous and continuous inward rolling at a speed of $3.9° \, s^{-1}$ and anticlockwise rotation at a speed of $1.3° \, s^{-1}$.

One remarkable observation is that the motion speeds of knotbots are directly correlated with the number of crossings. As shown in Fig. 4e, with the increasing number of crossings, the less effective segments between crossings impose tighter constraints, thus hampering the motion of knotbots. Interestingly, the ratio

between rotation speed $\dot{\theta}$ and rolling speed $\dot{\gamma}$ is solely determined by the structural topology−it is inversely proportional to the number of crossings (Fig. 4f). To reveal the kinematics, let's imagine the cross-sectional view through the cut plane A-A depicted in Fig. 4g. The cross-sections of the two strands are marked with different colors, and diameter lines are added for clearer presentation. If no sliding takes place between the two strands, the relative position of the cross-sections is solely determined by the average rolling angle $\gamma_{avg}$ (e.g., no change in relative position when the two roll in opposite directions by the same angle). A section is located at a crossing when the two strands are in a top-down setup. Due to the self-shadowing effect, the strand sitting on top of a crossing is heated up more and temporarily rolls faster. When the rolling motion switches the relative position of the two strands, i.e., when $\gamma_{avg} = \pi$ as illustrated by state 1 in Fig. 4g, the same location is now occupied by the next crossing, and the apparent braid rotation has traveled from one crossing to the next: $\theta = 2\pi/n$ (e.g., $2\pi/3$ for a trefoil). The process continues after state 1, but now the red section sits on top and thus rolls faster. It takes another half cycle of average rolling, when the rolling motion switches the relative locations of the two strands again, back into the initial state (state 2), $\gamma_{avg} = 2\pi$. The braid rotation has traveled through two crossings, with the apparent rotation angle $\theta = 4\pi/n$. As another example, a Solomon link rotates by $\theta = \pi$, when the relative rolling of the two strands in a section completes a full cycle (Fig. 4h). Therefore, the ratio between the braid rotation angle and the rolling angle equals approximately $2/n$, corresponding to a slop of 2 in the linear correlation between $\dot{\theta}/\dot{\gamma}$ and $1/n$, which is consistent with the experimental observation.

The knotbots can be designed to operate simple machines or carry out certain tasks. As shown in Fig. 5a and Supplementary Movie 7, a gear is placed atop a right-handed trefoil knotbot. Upon uniform light irradiation, the continuous rolling of the trefoil knotbot rotates the gear by friction. After light irradiation for 35 s, the gear is rotated by 220°. By harnessing the rolling motion, the trefoil knotbot can climb a threaded rod at a speed of $0.3 \, mm \, s^{-1}$ under uniform light irradiation (Fig. 5b and Supplementary Movie 7). As expected, the climbing motion can be steered bidirectionally. When irradiated from the bottom, the knotbot climbs downward. A similar climbing motion of the trefoil knotbot is also achieved upon scanning-light irradiation (Supplementary Fig. 26). Evidently, the climbing is driven by the rolling motion of the knotted hydrogel and influenced by the friction between the rod and the gel. The trefoil knotbot can even climb along a smooth rod with comparable diameter, while simultaneously lifting and rotating a gear with the speed of $0.2 \, mm \, s^{-1}$ and $2.5° \, s^{-1}$, respectively (Fig. 5c and Supplementary Movie 7). The continuous motion of the knotbot can also be harnessed to transport a payload along a horizontal string. As shown in Fig. 5d and Supplementary Movie 8, the rolling of the two strands leads to horizontal locomotion of the trefoil knotbot, which carries a ring along at a speed of $0.46 \, mm \, s^{-1}$. Interestingly, the rolling of gel strands in the knotbot also results in the rotation of the plastic ring at a speed of $5.5° \, s^{-1}$. Besides these demonstrations, autonomous knotbots should find other applications in biomedical and engineering fields, especially when controlled stimulations are inaccessible.

**Fig. 4 | Continuous self-regulated motion of diverse knotbots. a–c** Snapshots of the anticlockwise braid rotation and inward rolling of a right-handed pentafoil (**a**), Solomon-link (**b**), and Star-of-David link (**c**) knotbots under uniform light irradiation. **d** Variations of rotation angle $\theta$ and rolling angle $\gamma$ of the knotbots with time under uniform light irradiation. **e** Rotation speed $\dot{\theta}$ and rolling speed $\dot{\gamma}$ of knotbots with different topology. **f** The relation between the ratio of $\dot{\theta}$ to $\dot{\gamma}$ and the number of crossings of the knotbots with different circumference $L$ of gel strand under uniform light irradiation of different power intensity $I$. **g**, **h** Schematic to show the coupled rolling and braid rotation of contacting strands without sliding, in trefoil (**g**) and Solomon-link (**h**) knotbots. Data points are represented as mean ± s.d. ($n \geq 3$).

## Discussion

Inspired by the principle of symmetry-refresh-mediated locomotion of vibrant biological organisms, we have proposed a specific type of knotted soft robots with embedded asymmetric structures and topological constraints, the knotbots, which exhibit autonomous motions under static or dynamic light irradiation. Owing to the interplay between the self-constraint-induced prestrains and the photothermal strains, the photo-responsive knotbots elicit self-regulated dynamic motions. We have demonstrated that the knotbots in various forms, ranging from the torus to trefoil/pentafoil knots and complex

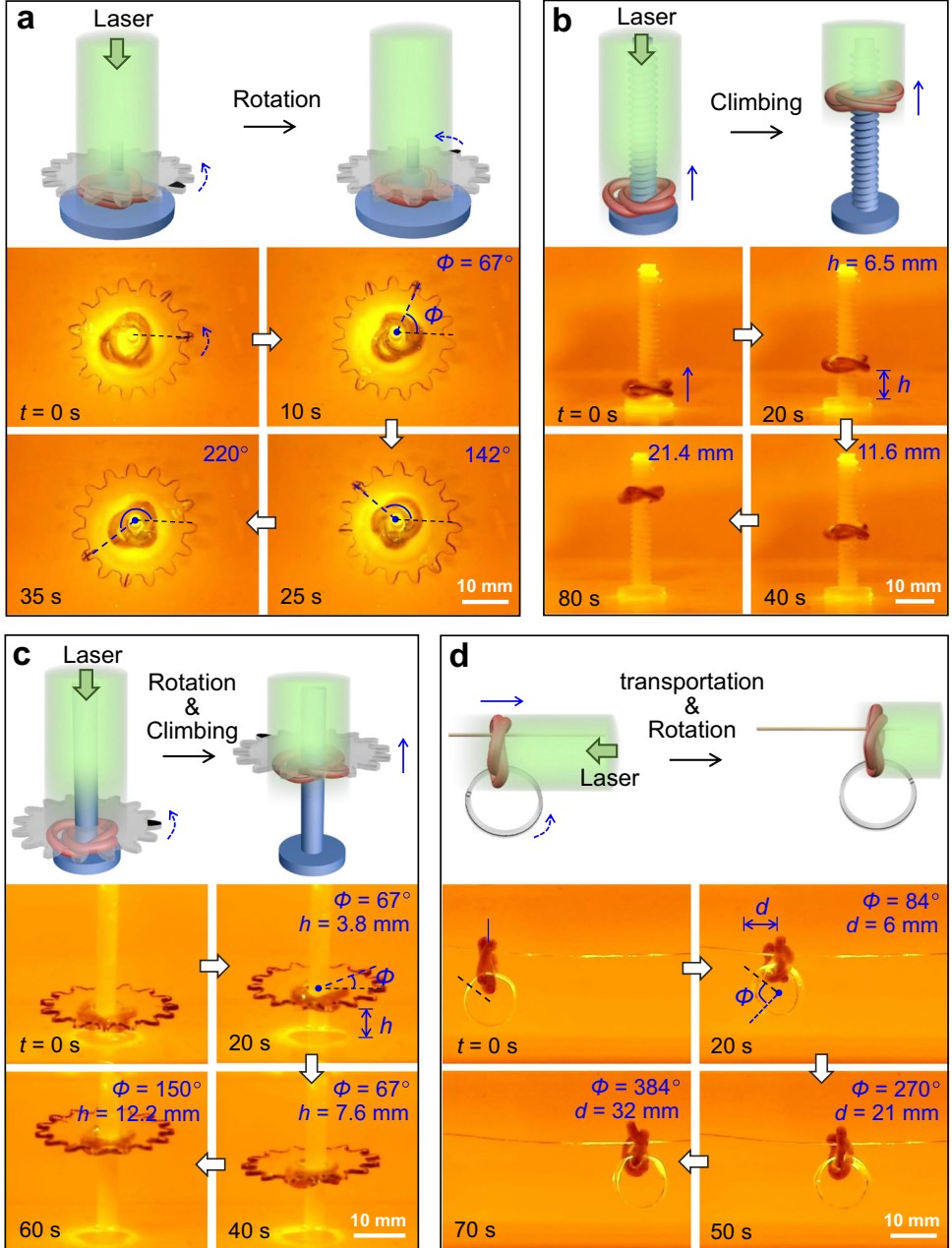

**Fig. 5 | Locomotion of knotbot for various sample tasks. a** A trefoil knotbot rotating a gear under uniform light irradiation. **b** A trefoil knotbot climbing along a vertical threaded rod under uniform light irradiation. **c** A trefoil knotbot climbs along a smooth rod to lift and rotate a gear under uniform light irradiation. **d** Transportation of a trefoil knotbot along a horizontal string under uniform light irradiation. The rolling of the knotbot leads to horizontal locomotion as well as the rotation of an interlocked ring.

Solomon/Star-of-David links, exhibit rolling, spinning, and braid rotation under light irradiation. This result indicates that the topology-induced constraints and prestrains of knotbots afford the responsive hydrogels with programmed actuation, physical intelligence, and collective motions by symmetry breaking and self-shadowing effect. In terms of symmetry refresh, our knotbots are quite similar to internal combustion engines (ICEs), in that both use heat to break symmetry and rolling to restore it. However, in an ICE, multiple cylinders working in parallel through out-of-phase cycles are often linked by the crankshaft, to induce a discrete self-similarity with a countable degree of freedom[38]. Whereas, every cross-section in the knotbot is capable of producing torque and contributes to the overall motion. Featured with infinite degrees of freedom through closed loop and continuum deformation, knotbot doesn't require any extra self-regulated component and is natural for smooth and continuous motion. This intelligence arising from the topology-invoked dynamic actuation and self-regulation of active materials enables the self-sustained motion of soft robots. This work calls for rethinking the topology of soft robots and shifting the spotlight from computational intelligence to physical intelligence[39,40]. It remains an open question on which knotted structures can afford self-regulation and be applicable to devise autonomous soft robots. Another promising direction is the transformative applications of these knotbots in the biomedical field, for which they should be miniaturized and fabricated through advanced manufacturing technologies. The design principle and kinematics of knotbots should be applicable to other responsive materials[40], and should open exciting perspectives to soft machines.

## Methods

### Synthesis of hydrogels

The NS powders were added to specific amount of deionized water for one month at room temperature, and then the homogeneous suspensions of 0.5 wt%, 1.0 wt%, 1.5 wt%, and 2.0 wt% were obtained. NIPAm of 10.1 wt%, *N,N*′-methylenebis(acrylamide) (MBAA, chemical crosslinker) of 0.275 wt%, lithium phenyl-2,4,6-trimethylbenzoylphosphinate (LAP, photo-initiator) of 0.106 wt%, and AuNPs of 0.25 wt% or 0.38 wt% were added into the above NS suspension to prepare the precursor solution. The precursor solution was injected into a transparent rubber tube with an inner diameter of 2.0 mm through a syringe at a specific flow rate. The ordered structure of NSs in the precursor solution was permanently fixed by immediate ultraviolet (UV) light irradiation for 100 s to accomplish the photopolymerization, and the anisotropic cylindrical hydrogels were obtained. Then, the cylindrical hydrogels were equilibrated in a large amount of deionized water to equilibrate. Details about the synthesis of NSs and AuNPs are described in the Supplementary Information.

### Fabrication of knotbots

The cylindrical hydrogels with predetermined lengths were twined and joined at the ends with a soft glue to obtain the knotbots, including torus, single prime knots (e.g., trefoil and pentafoil knots), and complex links (e.g., Solomon and Star-of-David links). The joint region had little effect on the continuous deformation and motion of the knotbots. Internal chirality (i.e., torsional prestrain) was introduced into the knotbots by twisting the cylindrical gel clockwise or anticlockwise with a prescribed twist number, which was maintained in the knot after joining the two ends of the gel strand. For example, a gel twisted clockwise by 360° and then joined at the ends to form a torus with right-handed internal chirality is noted as the $T_{+1}$ knotbot. For precise control of the twist number, a black line was drawn on the cylindrical gel as the marker.

### Animating of knotbnots

Motions of the knotbots were animated upon static or dynamic light stimulations. The knotbot was placed atop a flat polyvinyl chloride substrate and incubated in a water bath of a certain temperature that was controlled by a heating-cooling system and monitored with a thermometer. Under uniform light irradiation (wavelength, 520 nm), the knotbot exhibited dynamic morphing and motions that were recorded as movies by a digital camera equipped with a cut-off filter (550−1100 nm) to filter out the green light. The light intensity was controlled by adjusting the power of the laser and the distance between the laser and the gel. Upon scanning-light irradiation, the knotbots were animated in a similar way. For example, the gel torus was animated by the scanning of a light spot (diameter, 10 mm) along the circumference of the gel with a constant speed of 5 mm s⁻¹. For the motions of knotbots under dynamic light irradiation, the temperature of the water bath was kept as 19 °C.

### Characterizations

Characterizations of microstructures and properties of the NSs, AuNPs, cylindrical gels, and knotbots are described in the Supplementary Information.

### Multiphysics modeling

**Theoretical model.** The dynamic deformation and locomotion of the knotbots are modeled by solving coupled fields of hyperelastic deformation, thermal radiation, and frictional contact. Following the model of a similar material system[25], the phase-transition-induced isochoric deformation is modeled by introducing an inelastic deformation gradient

$$\mathbf{F}^{(i)} = \frac{1}{\sqrt{\lambda_n}}\mathbf{e}_{\parallel}\otimes\mathbf{e}_{\parallel} + \frac{1}{\sqrt{\lambda_n}}\mathbf{e}_{\perp}\otimes\mathbf{e}_{\perp} + \lambda_n\mathbf{e}_n\otimes\mathbf{e}_n, \tag{1}$$

where $\mathbf{e}_{\parallel}$ and $\mathbf{e}_{\perp}$ are the unit vectors parallel and perpendicular to the shear-flow direction during precursor preparation, both within the plane of the NSs, $\mathbf{e}_n$ is the unit vector normal to the NS plane, $\lambda_n$ is the stretch ratio along $\mathbf{e}_n$, and the operator $\otimes$ denotes a tensor product. Knowing that the phase-transition-induced deformation is fast relative to the heating rate, it is further assumed that $\lambda_n$ is a function of the normalized temperature $T$, which ramps linearly from $T = 0$ (room temperature) to $T = 1$ at which the isochoric deformation saturates.

To describe the deformation, we take the state prior to the isochoric deformation ($T = 0$) as the reference and employ the multiplicative decomposition. Specifically, we write the deformation gradient tensor as the inner product between the inelastic and elastic deformation gradients

$$\mathbf{F} = \mathbf{F}^{(e)} \cdot \mathbf{F}^{(i)}. \tag{2}$$

To characterize the elastic response of the material, we introduce the Helmholtz free energy density $W$, which is taken to be a function of the elastic deformation gradient $\mathbf{F}^{(e)}$. The processes of interest are dominated by the isochoric deformation of the gel (Supplementary Fig. 7), and are relatively insensitive to its anisotropic elastic properties (Supplementary Fig. 11). For simplicity, we adopt a neo-Hookean model with isotropic initial shear modulus $G$

$$W\left(\mathbf{F}^{(e)}\right) = \frac{G}{2}\left(\mathbf{F}^{(e)} : \mathbf{F}^{(e)} - 3\right), \tag{3}$$

and assume volume incompressibility. The equation of state can then be given in terms of the nominal stress

$$s_{iK} = \frac{\partial W}{\partial F_{iK}} = \frac{\partial W}{\partial F_{iJ}^{(e)}}L_{KJ}^{(i)}, \tag{4}$$

where $\mathbf{L}^{(i)}$ is the inverse of the inelastic deformation gradient tensor $\mathbf{F}^{(i)}$. The nominal stress $\boldsymbol{s}$ equilibrate with the distributed body force $\mathbf{b}$ as

$$\nabla \cdot \mathbf{s} + \mathbf{b} = \mathbf{0} \tag{5}$$

in the bulk, and boundary condition $\boldsymbol{s} \cdot \mathbf{N} = \mathbf{t}$ on a surface of prescribed nominal traction $\mathbf{t}$, with $\mathbf{N}$ being the unit normal vector of the surface in the reference state. As the motion of the gels is relatively slow, all inertial effects are neglected in the model, except for the gravity which is necessary for the frictional contact between a knotbot and the supporting surface.

When heated under a laser beam, the region of elevated temperature is usually localized with a minor effect of thermal conduction. We thus neglect thermal conduction and simplify the temperature evolution into a modified Stefan−Boltzmann equation

$$\frac{\partial T(\mathbf{X}, t)}{\partial t} = r(\mathbf{x}(\mathbf{X}), t) - \beta T(\mathbf{X}, t). \tag{6}$$

Here $r(\mathbf{x}, t)$ characterizes the heating intensity of the laser beam, which is localized at the irradiated region and decays with depth, depending on the current (deformed) position $\mathbf{x}$, and $\beta$ is a parameter characterizing the emissivity of the gel to the environment. The

parameters are obtained by fitting the dynamic response of a mono-domain gel (Fig. 1h).

The frictional contact is modeled by applying a penalty contact force in the normal direction

$$f_n = \begin{cases} -d\exp(-d), & d < 0 \\ 0, & d \geq 0 \end{cases} \tag{7}$$

and a friction force tangential to the surface of the contact

$$f_t = -\eta f_n \frac{\mathbf{v}}{|\mathbf{v}|}, \tag{8}$$

where $d$ is the normal distance between the two surfaces, $\mathbf{v}$ is the relative sliding velocity, and $\eta$ is the friction coefficient.

**Finite element method implementation.** Upon substitution of Eqs. (1), (4), (7), and (8), Eqs. (5) and (6) are solved numerically by using a finite element method through the commercial package COMSOL Multiphysics 5.4. Considering the strong coupling, we solve the non-linear equations numerically with a fully coupled implicit method[41], and adopt automatic damping to stabilize the contact. Three-dimensional models representing the initial straight geometries of the samples, prior to knotting, are developed. As shown in Supplementary Fig. 17a, quadratic wedge elements are used to interpolate the field of displacement to capture the bending deformation precisely, and the normalized temperature is discretized over the Gauss points.

Modeling the special initial geometry of a knotbot, with properly prescribed self-constraints, contact, and prestrain, is challenging. Our approach is to take the straight state of a cylindrical gel strand at room temperature to be the reference state as well as the starting point (Supplementary Fig. 17). The cylinder is first twisted to control the torsional strain after connection (Supplementary Fig. 17c), and then deformed into the specific knot shape without contact constraint, gravity, or rolling[37] (Fig. 1d-f). Overlap is allowed in the step. Afterward, the gravity and contact reaction forces are gradually added to naturally relax the knotbot into its initial state with proper topology and pre-strain. Additionally, a periodic boundary is prescribed at the two ends to form a closed loop.

**Additional hoop stress profile and modes of motion**
To reveal the underlying mechanism of each mode of motion, we extract the additional hoop stress due to light irradiation. It is note-worthy that the light-induced stress is small relative to the prestress induced during the knotting process. As shown by Supplementary Fig. 14a, b, the difference in hoop stress before and after several sec-onds of light irradiation is almost imperceptible in a rolling knotbot, aside from the displacement caused by rolling. Moreover, the large deformation is strongly coupled with the heating process, e.g., the rolling tends to switch the region of direct irradiation. Therefore, we intentionally stall all material particles in a rolling knotbot, while continuing the heating processing. Such a state differs from that of a rolling knotbot, but exaggerates the driving force for the impending motion. We calculate the hoop stress in the state after the stalled irradiation (Supplementary Fig. 14c), deduct that in the state prior to irradiation (Supplementary Fig. 14b), and plot the differential hoop stress profile in Supplementary Fig. 14d.

Clearly, the light irradiation induces excess tensile hoop stress in the top half of the knotbot and compression in the bottom half (Fig. 2d). Considering a segment of the torus that is being irradiated, the tensile and compressive hoop stresses on the cross-sections result in concentric and eccentric forces, respectively. The separated lines of action between these resultants induce a couple moment, which drives the inward rolling of the torus as illustrated in Fig. 2b.

Similarly, the excessive hoop stress in a trefoil knotbot is also shown in Fig. 3d, and the simulated results that polar angle $\varphi$ and polar radius $\rho$ of a specific cross-section, as well as the rolling angle $\gamma$ of a specific particle are also extracted as functions of time (Fig. 3f). In the cross-sectional view depicted in Fig. 3d, the tensile and compressive hoop stresses in a single strand also induce the inward rolling akin to that in a torus. When the entire cross-section of both strands is taken into account, the same mechanism drives the rotation of the entire braid, thus manifesting the braid rotation. This braid rotation is mea-sured in simulation by tracking the rotation of the connecting line between the centers of the two strands in a cross-section. On the other hand, the spinning motion is attributed to friction between the knot-bot and the supporting surface, and is expected to diminish on a frictionless surface lacking tangential forces that can provide the torque.

**Kinematics of torus knotbots with pre-torsion under scanning light**
The numerical results of three torus knotbots with different pre-torsion, namely the $T_0$, $T_{+1}$, and $T_{-1}$ knotbots, are plotted in Fig. 2j-l. Three equally spaced cross-sections are marked in white on each knotbot, with their relative angles of twist indicated. For example, the original angles of twist prior to light irradiation are 14.6° and −14.6° over the first and the third sections in a $T_{+1}$ knotbot, relative to the middle section, while no original torsion exists in a $T_0$ knotbot.

When a knotbot is subject to light irradiation, additional torsion will be induced (e.g., in Fig. 2j-l). In a $T_0$ knotbot subject to stationary irradiation on top of the middle cross-section, the angles of twist are both −3.4° on the other sections. In contrast, when the same $T_0$ knotbot is subject to anticlockwise scanning-light irradiation, the relative angles of twist are −2.1° and −5.1° over the first and third sec-tions, respectively, at the moment when the irradiated spot is on top of the second section. The asymmetry is caused by the residual heat behind the irradiated spot. In a $T_{+1}$ knotbot, due to the pre-torsion, the additional angles of twist are −10.3° and 9.9° under anticlockwise scanning and 6.3° and −7.2° under clockwise scanning. Such asym-metry in photothermal torsion drives the anchor point away from the irradiated spot, rendering a stronger driving moment for faster rolling and spinning.

## Data availability
All the data that support the findings of the study are available within the article and its Supplementary Information. Extra data are available from the corresponding authors upon request. Source data are pro-vided in this paper.

## Code availability
A simulation code in the study will be provided upon request.

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

## Acknowledgements

We acknowledge funding support from National Natural Science Foundation of China (52325302, 51973189, 52173012, Z.L.W.; 11972015, H.W.), Shanxi-Zheda Institute of Advanced Materials and Chemical Engineering (2022SZ-FR004, Z.L.W.), and the Science, Technology, and Innovation Commission of Shenzhen Municipality (ZDSYS20210623092005017, W.H.). We thank BL 16B1 beamline of Shanghai Synchrotron Radiation Facility (SSRF) for SAXS measurement.

## Author contributions

Q.L.Z. and Z.L.W. convinced the concept and designed the experiments. Q.L.Z. performed the experiments with the assistance of O.K. and J.B. for the synthesis of nanosheets. W.L. and W.H. developed the theoretical model and carried out the numerical simulations to elucidate the kinematic mechanisms. All the authors contributed to the discussion of the results and writing of the manuscript. W.H., Q.Z., and Z.L.W. supervised the project.

## Competing interests

The authors declare no competing interests.
