## [Peer Review File · Nature Communications]

Animating hydrogel knotbots with topology-invoked self-regulationREVIEWER COMMENTS

Reviewer #1 (Remarks to the Author):

This is a truly elegant and novel study. The researchers describe how the knotted geometry induces constraints on the motion of the knotbot gels and how these constraints allow the material to undergo regular, self-regulated motion. To the the best of this reviewer's knowledge, there has not been much published on the dynamic, self-regulating behavior of soft, knotted materials. The work include experiments and modeling, which concurs with the experimental findings.

This paper will be of significant interest to the soft active matter community. Its novelty and careful analysis make this paper worthy of publication in Nat. Comm. and can be published as is. It is a very timely study and it would be good to get it out in the community.

Reviewer #2 (Remarks to the Author):

The work presents an interesting idea for obtaining motion in soft robots. Despite morphology being an important aspect of soft robot design, here topology and symmetry breaking have been developed to a very good level. The paper is well-written and clear. However, the presentation of some more introductory concepts, for readers that are not fully in the field, would improve overall reading. The results show robust and reproducible motions. A few different designs and morphologies have been tested. The question is open on how many more possible configurations would be effective and how to design them.

The videos show possible applications in basic locomotion movements, as well as in basic mechanical motions. As a minor remark, the videos show very long titles, in a time that is too short to read them. They are in fact long and helpful captions that may be used in that way instead, e.g., left there for the whole video duration.

Reviewer #3 (Remarks to the Author):

the paper presents a novel concept of knotbots with self-regulated motion, backed by a well-documented experimental study. To enhance its impact, the paper could further discuss the practical applications and future directions of this research in the field of soft robotics. The study represents innovative research in the field of soft robotics, introducing the concept of knotbots with self-regulated motion. This novel approach opens up exciting possibilities for the development of intelligent soft robots. Additionally, some minor improvements in connecting knot theory to robotics in the introduction could enhance the paper's clarity.

[+] The paper draws parallels with biological systems, such as molecular motors and organisms with cooperative motions. While these analogies can be insightful, it's important to acknowledge the fundamental differences between biological systems and artificial soft robots. A more critical discussion of the limitations of such analogies would strengthen the paper.

[+] The paper would benefit from a more detailed discussion of the control variables in the experiments. For example, how were the variables like light intensity and temperature precisely controlled and measured? Any variations or uncertainties in these parameters can affect the repeatability of the results.

[+] If theoretical models were used to explain the observed phenomena, it is essential to discuss how well these models align with the experimental results. Were there any deviations or limitations in the models' predictions compared to the actual behavior of the knotbots?

[+] The paper discusses the motion of knotbots but doesn't delve into potential practical applications in detail. A more in-depth exploration of how this research could be applied in fields like medicine, manufacturing, or microsurgery would enhance the paper's relevance. The practical applications of these knotbots are not extensively explored in the study. While the research showcases intriguing capabilities, the transition from the laboratory to real-world applications may be challenging and could face unanticipated limitations.

[+] The paper mentions the use of specific materials like fluorohectorite nanosheets and gold nanoparticles. While it provides some characterization, a more comprehensive discussion of the material properties, such as size distribution, stability, and any potential limitations, would be valuable.

[+] The paper mentions the importance of prestrain in achieving motion but doesn't elaborate on how prestrain was precisely controlled or measured in the hydrogel. Providing more details on this aspect would aid in understanding the mechanical behavior of the knotbots.

[+] The motion of knotbots is sensitive to various experimental conditions such as light intensity, temperature, and geometry. These sensitivities could limit the reproducibility and reliability of the observed behavior. The authors are suggested to provide discuss these sensitivities.

[+] The choice of materials, such as fluorohectorite nanosheets and gold nanoparticles, may not be easily scalable or readily available for all applications. This could limit the practicality of replicating the experiments. The authors are suggested to provide some comments pertaining this issue.

[+] The study explores several knot topologies but is not exhaustive in this regard. There are numerous knot topologies, and each may have unique mechanical behaviors. The generalizability of the findings to other knot types is uncertain.

In conclusion, the study on hydrogel-based knotbots with self-regulated motion is a contribution to the field of soft robotics. Its innovative approach and potential for transformative applications make it a commendable piece of research.

Response to Reviewers' Comments

We sincerely thank the reviewers for their time and thoughtful comments on our manuscript (NCOMMS-23-39980-T). We have carefully revised the manuscript in response to their questions and suggestions. In the following, the reviewers' comments appear in black, and our itemized responses are in blue. The revisions made to the manuscript are marked in red.

Reviewer #1 (Remarks to the Author):

This is a truly elegant and novel study. The researchers describe how the knotted geometry induces constraints on the motion of the knotbot gels and how these constraints allow the material to undergo regular, self-regulated motion. To the the best of this reviewer's knowledge, there has not been much published on the dynamic, self-regulating behavior of soft, knotted materials. The work includes experiments and modeling, which concurs with the experimental findings.

This paper will be of significant interest to the soft active matter community. Its novelty and careful analysis make this paper worthy of publication in Nat. Comm. and can be published as is. It is a very timely study and it would be good to get it out in the community.

Response: Thank you very much for the high evaluation and recommendation for publication of our manuscript. We are delighted that you found our work elegant and novel.

Reviewer #2 (Remarks to the Author):

The work presents an interesting idea for obtaining motion in soft robots. Despite morphology being an important aspect of soft robot design, here topology and symmetry breaking have been developed to a very good level. The paper is well-written and clear. However, the presentation of some more introductory concepts, for readers that are not fully in the field, would improve overall reading.

Response: We sincerely appreciate for your positive evaluation and helpful comments on our manuscript. We understand the importance of clear and comprehensible presentation, particularly for those who may not be deeply immersed in the field. We have revised the manuscript to improve the overall readability and accessibility. We have added more details about the definition of knots and braid rotation (pages 3 and 12), as well as the introduction to conceptual terms of physical intelligence and hoop stress (pages 4 and 8).

The results show robust and reproducible motions. A few different designs and morphologies have been tested. The question is open on how many more possible configurations would be effective and how to design them.

Response: We totally agree with that it is an open and intriguing area for future research. There is a large number of different types of knots, according to the knot theory, and it is impossible to exhaust all based on experiments and simulations. In the current work, we have selected several typical knots, which exhibit self-regulated, continuous motions under uniform light irradiation. The main purpose is to emphasize the importance of topology in the design of soft robots, which invokes self-regulation for the autonomous motions. Even though such a practice does not answer the ultimate question of design, we believe that the principle and kinematics revealed in this work are universal and should facilitate further research on topology-based soft robots. We have added this open question to the manuscript to inspire readers to explore other knotbots (page 22).

The videos show possible applications in basic locomotion movements, as well as in basic mechanical motions. As a minor remark, the videos show very long titles, in a time that is too short to read them. They are in fact long and helpful captions that may be used in that way instead, e.g., left there for the whole video duration.

Response: Many thanks for this helpful suggestion. We have extended the duration of display for the titles to ensure readability.

Reviewer #3 (Remarks to the Author):

The paper presents a novel concept of knotbots with self-regulated motion, backed by a well-documented experimental study. To enhance its impact, the paper could further discuss the practical applications and future directions of this research in the field of soft robotics. The study represents innovative research in the field of soft robotics, introducing the concept of knotbots with self-regulated motion. This novel approach opens up exciting possibilities for the development of intelligent soft robots. Additionally, some minor improvements in connecting knot theory to robotics in the introduction could enhance the paper's clarity.

Response: Thank you very much for the positive feedback and valuable suggestions. Accordingly, we have revised the manuscript to discuss the applications and future directions in the Discussion part (page 22). Regarding the knot theory, it is, in mathematics, the study of closed loops in 3D space, with Alexander-Briggs notation to classify different knots and link. For example, trefoil knots and Solomon link are noted as 3_1 and 4_1^2 respectively. In our work, we have not used this notation for easy reading. We have emphasized the significances of knotted structures for the robotic functions, especially the self-regulation, of the soft robots: “owing to the interlacement of closed loops, knots often include topological constraints, bring self-shading and prestress, and thus may enhance actuation performance and afford physical intelligence” (page 4).

[+] The paper draws parallels with biological systems, such as molecular motors and organisms with cooperative motions. While these analogies can be insightful, it's important to

acknowledge the fundamental differences between biological systems and artificial soft robots. A more critical discussion of the limitations of such analogies would strengthen the paper.

Response: Biological systems usually have complex musculature and neural networks with intricate responses to bio-signals, which afford high-efficiency sensing, actuation, control, and motions. Especially, the neural intelligence facilitates self-sustained, autonomous motions by self-regulated shape morphing. In soft robots, it is hard to replicate such neural intelligence. Rather, physical intelligence, which has also been harnessed by biological systems based on unique structures and topologies, appears to be a promising way to realize self-regulated, autonomous motions of soft robots. The material systems and biotissues are soft and responsive to stimulations, sharing some similarities. Therefore, we introduce biological and artificial systems by analogy, to emphasize the self-regulation capacities based on with internal and/or external feedback loops. Such physical intelligence has not been well studied in artificial soft robots, especially considering the topology of responsive materials.

We totally agree that the analogies between the biological systems and artificial soft robots in our manuscript have some limitations. According to your suggestion, we have discussed these limitations in the revised manuscript (page 3).

[+] The paper would benefit from a more detailed discussion of the control variables in the experiments. For example, how were the variables like light intensity and temperature precisely controlled and measured? Any variations or uncertainties in these parameters can affect the repeatability of the results.

Response: Variations in key parameters can significantly affect the motions of knots. In our experiments, we have meticulously managed these factors to ensure repeatability and reliability. According to your suggestion, we have incorporated extensive discussion on the detailed control and measurements of these variables. The light intensity is controlled by adjusting the power of the laser and the distance between the laser and the gel; the temperature of the water bath is controlled by a heating-cooling system and monitored with a thermometer.

The phase diagrams of the motion states of the knotbots in Fig. 2e and 3i indicate that the knots can maintain continuous motions over a wide range of temperature and light intensity. These robust motions manifest the reliability of our experimental results. The data are obtained from three or more parallel experiments, which further underscore the high level of repeatability and reliability. We have clarified this point in the revised manuscript (page 11).

[+] If theoretical models were used to explain the observed phenomena, it is essential to discuss how well these models align with the experimental results. Were there any deviations or limitations in the models' predictions compared to the actual behavior of the knotbots?

Response: We appreciate the insightful comments on the alignment of theoretical models with experimental results. While we acknowledge the differences between the computational models and the real situation, it's crucial to highlight that these differences can largely be attributed to simplifying assumptions made for the sake of computational tractability. We have endeavored

to ensure that our models align with experimental results through the following assumptions. (i) In the experimental setup, it is observed that the light intensity decays non-linearly within the gel. However, as the detailed distribution of light intensity is unknown and not deemed as an important factor as suggested by our numerical tests, a heat source field that decays linearly with depth is used for simplicity in the simulations. (ii) In reality, the friction of the gel against the substrate and that between two parts of a gel should both be functions of the relative velocity. To facilitate computation while retaining a reasonable level of fidelity, we employ constant friction coefficients for both cases. These approximations and simplifications have proven effective in ensuring a close match with the experimental results.

While reasonable assumptions make the simulations align with the experimental results, it is also important to acknowledge certain limitations in specific situations. (i) In the cases of significant reflection or refraction of light, a pre-defined temperature field may be inadequate. A detailed model of the light intensity distribution can be incorporated for more accurate representation. (ii) If thermal conduction is significant, our simple treatment of taking thermal radiation as the only means of heat dissipation may be less accurate, and a more complete model including heat conduction may be needed. The extensions, however, are deemed as unnecessary in the current study.

Corresponding revisions and discussions on this point have been made to the theoretical modeling part of the manuscript (page 11 in the manuscript; pages 3 and 4 in the Supplementary Information).

[+] The paper discusses the motion of knotbots but doesn't delve into potential practical applications in detail. A more in-depth exploration of how this research could be applied in fields like medicine, manufacturing, or microsurgery would enhance the paper's relevance. The practical applications of these knotbots are not extensively explored in the study. While the research showcases intriguing capabilities, the transition from the laboratory to real-world applications may be challenging and could face unanticipated limitations.

Response: Thank you very much for this critical comment. As you pointed out, exploring real-world applications should further strength the significance and impacts of our work. In this study, our primary focus is the new design principle of soft robots with topology-invoked self-regulation and the underlying mechanism of autonomous motions. This work may pave the way to devise novel soft robots capable of sustainable motions under constant conditions. Considering the complex situations in biomedicines, self-regulated motions should have particular significances, especially when dynamic stimulations are not accessible. For these bio-related applications, the knotbots should be miniaturized and engineered through other advanced manufacturing technologies. This work is ongoing in our lab. In the revised manuscript, we have simply discussed the advantages and challenges of real-world applications of the knotbots (pages 21 and 22).

[+] The paper mentions the use of specific materials like fluorohectorite nanosheets and gold nanoparticles. While it provides some characterization, a more comprehensive discussion of the material properties, such as size distribution, stability, and any potential limitations, would be valuable.

Response: The fluorohectorite $[\text{Na}_{0.5}][\text{Li}_{0.5}\text{Mg}_{2.5}][\text{Si}_4]\text{O}_{10}\text{F}_2$ nanosheets used in this work have a large aspect ratio of $\sim 20,000$ (average size of $\sim 20 \mu\text{m}$, thickness of $\sim 1 \text{ nm}$) and high charge density of 1.1 nm^{-2} , which are critical for the electrical orientation of the nanosheets and the permittivity-mediated anisotropic deformation of the gel. To show the general structure of the nanosheets, an atomic force microscope (AFM) image is presented in Supplementary Fig. 1. More detailed characterizations of the nanosheets have been reported in the literatures (*Langmuir* 32, 10582-10588 (2016); *ACS Appl. Mater. Interfaces* 5, 5851-5855 (2013)) by Prof. Josef (a co-author of the current work). The large aspect ratio of the nanosheets and the directional alignment induce the anisotropic mechanical properties of the composite - stiff in the in-plane direction and flexible through-thickness. The nanosheet suspensions used in this work are obtained by delamination of the fluorohectorite powders into single lamellae in water through osmotic swelling. The nanosheet suspension forms homogeneous liquid crystalline phase at a low content (0.3 wt%), which is very stable at room temperature. Regarding the gold nanoparticles used in this study, the synthesis protocol is described in Methods of the manuscript. The synthesis procedure is reported in the literature (*Nat. Phys. Sci.* 241, 20-22 (1973)) and widely used by other researchers as high-efficiency photothermal agents. The obtained gold nanoparticles have an average diameter of $9.5 \pm 0.7 \text{ nm}$, as characterized by TEM (Supplementary Fig. 9a). Aqueous suspension of the gold nanoparticles has an absorbance peak at 520 nm (Supplementary Fig. 9b). The gold nanoparticles have a high stability after being incorporated into the anisotropic gel, which afford response to light. According to your suggestion, this information about the nanosheets and nanoparticles has been added to Supplementary Information (page 2).

[+] The paper mentions the importance of prestrain in achieving motion but doesn't elaborate on how prestrain was precisely controlled or measured in the hydrogel. Providing more details on this aspect would aid in understanding the mechanical behavior of the knotbots.

Response: Many thanks for the constructive suggestion. Prestrain is important for the continuous motions of the knotbots. There are two kinds of prestains in this study. When a straight cylindrical gel is bent into a torus or other knotted structures, bending and stretching prestains are introduced in the knotbots naturally. In addition, torsional prestrain is intentionally introduced into the torus gel or other knotbots by twisting the cylindrical gel clockwise or anticlockwise with a prescribed twist number, which is embedded in the knotbot by joining the two ends of the cylinder gel. For precise control of the twist number, a black line is drawn on the cylinder gel as the marker. The direct measurement of local prestrain within the knots is challenging, and we carry out finite element calculations to determine the prestains, similar as those in the literature (*Phys. Rev. Lett.* 99, 164301 (2007); *Extreme Mech. Lett.* 43, 101172 (2021)). According to your kind suggestion, the control of torsional prestrain by twisting the cylindrical gel is described in the Methods of the revised manuscript (page 23).

[+] The motion of knotbots is sensitive to various experimental conditions such as light intensity, temperature, and geometry. These sensitivities could limit the reproducibility and reliability of the observed behavior. The authors are suggested to provide discuss these sensitivities.

Response: Light intensity, environment temperature, and geometry are important factors that

affect the motion of the knotbots. We have systematically investigated these influences and summarized the modes of motion of torus and trefoil knotbots in a water bath of different temperatures and under uniform light of different intensities (Fig. 2e and 3e). The knotbots exhibit continuous motions over a wide range, indicating the robustness of the autonomous motions. Regarding to the geometry of the gel, we have examined the photo-response and the motions of knotbots made of cylindrical gel with different diameter and length. The results are shown in Supplementary Fig. 10d and 14c. Different geometries influence the motion behaviors to some extent. As you pointed out, these sensitivities may reduce the reproducibility of motion behaviors of the knotbots. If the experimental conditions are well controlled, the motions of knotbots exhibit good repeatability and reliability, as confirmed by parallel experiments. In addition, the sensitivity to experimental conditions and gel geometries also provide diverse parameters to tune the motion behaviors of the knotbots. For example, the motion speed of trefoil knotbot can be well tuned by the light intensity (Fig. 3g). According to suggestion, we have discussed the sensitive influences of experimental conditions and gel geometries on the morphing and motion behaviors of the knotbots in the revised manuscript (pages 8 and 11).

[+] The choice of materials, such as fluorohectorite nanosheets and gold nanoparticles, may not be easily scalable or readily available for all applications. This could limit the practicality of replicating the experiments. The authors are suggested to provide some comments pertaining this issue.

Response: In this work, the design of knotbots shows good repeatability that favors real-world applications. The fluorohectorite nanosheets are obtained through melt synthesis and scalable up to several tens of grams per batch. The gold nanoparticles can also be synthesized in a large quantity according to a general protocol reported in the literature (*Nat. Phys. Sci.* 241, 20-22 (1973)), which has been widely used by other groups. In fact, both the fluorohectorite nanosheets and the gold nanoparticles are commercially available. In addition, the contents of nanosheets and gold nanoparticles used for gel synthesis are very low (ca. 1 wt% and 0.4 wt%, respectively), which benefits mass production of the knotbots. Therefore, the materials used in this study should favor the applications of knotbots. Corresponding revisions have been made to clarify the scalable synthesis of nanomaterials and the merit for applications in Supplementary Information (page 2).

[+] The study explores several knot topologies but is not exhaustive in this regard. There are numerous knot topologies, and each may have unique mechanical behaviors. The generalizability of the findings to other knot types is uncertain.

Response: In the field of knot theory, a large number of knots have been defined, and it is impossible to exhaust all. In this manuscript, the knotbot designs are based on several typical knots, and exhibit self-regulated, continuous rolling and rotation motions under uniform light irradiation. The main purpose of this work is to emphasize the significance of topology for the dynamic actuation and self-regulated motions of the hydrogel robots. It has been demonstrated that the knotted structure has invoked physical intelligence and thus self-sustained motions under a constant condition, which is hardly achieved in other systems. We believe that the principle and kinematics revealed in this work are suitable for many knotted structures, which

may facilitate the design of new knotbots. As you pointed out, different knots may have distinct mechanical behaviors. It still remains elusive that which knots can be used to devise soft robots with autonomous motions. We have added this open question to the manuscript to inspire readers to explore other topology-based soft robots (page 22).

In conclusion, the study on hydrogel-based knotbots with self-regulated motion is a contribution to the field of soft robotics. Its innovative approach and potential for transformative applications make it a commendable piece of research.

Response: Thanks a lot for your positive evaluation and valuable comments that are really helpful to improve our manuscript. We sincerely hope that our responses and revisions have addressed your concerns and also improved the quality of this work.

REVIEWERS' COMMENTS

Reviewer #2 (Remarks to the Author):

The revised paper addresses the comments done for improving clarity in the introduction and discussion of key concepts involved in the work. Overall the paper has improved and gives a valuable contribution to the field.

Reviewer #3 (Remarks to the Author):

The authors have incorporated all my comments, and I don't have any additional comments. The manuscript has been significantly improved.